# Evaluation of a city-wide school-located influenza vaccination program in Oakland, California, with respect to vaccination coverage, school absences, and laboratory-confirmed influenza: A matched cohort study

Jade Benjamin-Chung[1]*, Benjamin F. Arnold[1,2], Chris J. Kennedy[1], Kunal Mishra[1], Nolan Pokpongkiat[1], Anna Nguyen[1], Wendy Jilek[1], Kate Holbrook[3], Erica Pan[3,4], Pam D. Kirley[5], Tanya Libby[5], Alan E. Hubbard[1], Arthur Reingold[1], John M. Colford, Jr.[1]

1 Division of Epidemiology and Biostatistics, University of California, Berkeley, Berkeley, California, United States of America, 2 Francis I. Proctor Foundation, University of California, San Francisco, San Francisco, California, United States of America, 3 Division of Communicable Disease Control and Prevention, Alameda County Public Health Department, Oakland, California, United States of America, 4 Department of Pediatrics, Division of Infectious Diseases, University of California, San Francisco, San Francisco, California, United States of America, 5 California Emerging Infections Program, Oakland, California, United States of America

* jadebc@berkeley.edu

## Abstract

### Background

It is estimated that vaccinating 50%–70% of school-aged children for influenza can produce population-wide indirect effects. We evaluated a city-wide school-located influenza vaccination (SLIV) intervention that aimed to increase influenza vaccination coverage. The intervention was implemented in ≥95 preschools and elementary schools in northern California from 2014 to 2018. Using a matched cohort design, we estimated intervention impacts on student influenza vaccination coverage, school absenteeism, and community-wide indirect effects on laboratory-confirmed influenza hospitalizations.

### Methods and findings

We used a multivariate matching algorithm to identify a nearby comparison school district with pre-intervention characteristics similar to those of the intervention school district and matched schools in each district. To measure student influenza vaccination, we conducted cross-sectional surveys of student caregivers in 22 school pairs (2017 survey, $N = 6,070$; 2018 survey, $N = 6,507$). We estimated the incidence of laboratory-confirmed influenza hospitalization from 2011 to 2018 using surveillance data from school district zip codes. We analyzed student absenteeism data from 2011 to 2018 from each district ($N = 42,487,816$ student-days). To account for pre-intervention differences between districts, we estimated difference-in-differences (DID) in influenza hospitalization incidence and absenteeism rates using generalized linear and log-linear models with a population offset for incidence

**Data Availability Statement:** Data from the vaccine coverage survey are available at https://osf.io/c8xuq/. Absentee data cannot be shared publicly because of protections under the Family Educational Rights and Privacy Act. Researchers may apply for access to absentee data from Oakland Unified School District at https://www.ousd.org/Page/1016 and from West Contra Costa Unified School District at https://www.wccusd.net/page/545. Influenza hospitalization data may be requested from the California Emerging Infections Program (info@ceip.us; https://ceip.us/contact/).

**Funding:** This study was supported by a grant from the Flu Lab (https://theflulab.org/) to the University of California, Berkeley (Award number: 20142281; PI: AR). The funders had no role in study design, data collection and analysis, decision to publish, or preparation of the manuscript.

**Competing interests:** I have read the journal's policy and the authors of this manuscript have the following competing interests: The following authors report grant support from the Flu Lab to the University of California, Berkeley for the conduct of this research: JBC, BFA, KM, NP, AN, WJ, AEH, AR, JMC. EP reports grant support from the Shoo the Flu organization to the Alameda County Public Health Department. KH reports paid employment from the Shoo the Flu organization. CK reports paid employment from Kaiser Permanente Northern California, Division of Research. PDK and TL report grant support from the U.S. Centers for Disease Control and Prevention to the California Emerging Infections Program.

**Abbreviations:** ACIP, Advisory Committee on Immunization Practices; DID, difference-in-differences; IIV, inactivated influenza vaccine; LAIV, live attenuated influenza vaccine; OUSD, Oakland Unified School District; SLIV, school-located influenza vaccination; WCCUSD, West Contra Costa Unified School District.

outcomes. Prior to the SLIV intervention, the median household income was $51,849 in the intervention site and $61,596 in the comparison site. The population in each site was predominately white (41% in the intervention site, 48% in the comparison site) and/or of Hispanic or Latino ethnicity (26% in the intervention site, 33% in the comparison site). The number of students vaccinated by the SLIV intervention ranged from 7,502 to 10,106 (22%–28% of eligible students) each year. During the intervention, influenza vaccination coverage among elementary students was 53%–66% in the comparison district. Coverage was similar between the intervention and comparison districts in influenza seasons 2014–2015 and 2015–2016 and was significantly higher in the intervention site in seasons 2016–2017 (7%; 95% CI 4, 11; $p < 0.001$) and 2017–2018 (11%; 95% CI 7, 15; $p < 0.001$). During seasons when vaccination coverage was higher among intervention schools and the vaccine was moderately effective, there was evidence of statistically significant indirect effects: The DID in the incidence of influenza hospitalization per 100,000 in the intervention versus comparison site was −17 (95% CI −30, −4; $p = 0.008$) in 2016–2017 and −37 (95% CI −54, −19; $p < 0.001$) in 2017–2018 among non-elementary-school-aged individuals and −73 (95% CI −147, 1; $p = 0.054$) in 2016–2017 and −160 (95% CI −267, −53; $p = 0.004$) in 2017–2018 among adults 65 years or older. The DID in illness-related school absences per 100 school days during the influenza season was −0.63 (95% CI −1.14, −0.13; $p = 0.014$) in 2016–2017 and −0.80 (95% CI −1.28, −0.31; $p = 0.001$) in 2017–2018. Limitations of this study include the use of an observational design, which may be subject to unmeasured confounding, and caregiver-reported vaccination status, which is subject to poor recall and low response rates.

## Conclusions

A city-wide SLIV intervention in a large, diverse urban population was associated with a decrease in the incidence of laboratory-confirmed influenza hospitalization in all age groups and a decrease in illness-specific school absence rate among students in 2016–2017 and 2017–2018, seasons when the vaccine was moderately effective, suggesting that the intervention produced indirect effects. Our findings suggest that in populations with moderately high background levels of influenza vaccination coverage, SLIV programs are associated with further increases in coverage and reduced influenza across the community.

## Author summary

### Why was this study done?

- Seasonal influenza is a substantial contributor to hospitalization and mortality, particularly among infants and the elderly.

- Mathematical models project that vaccinating at least 80% of school-aged children, who are responsible for the majority of influenza transmission, may yield substantial community-wide reductions in influenza transmission through herd immunity.

- School-located influenza vaccination (SLIV) programs may increase influenza vaccination coverage among children. Prior studies of SLIV programs did not use designs that

effectively controlled for differences between schools with and without SLIV programs or were conducted in small numbers of schools.

## What did the researchers do and find?

- We conducted a 4-year evaluation of a large-scale SLIV program in ≥95 preschools and elementary schools in a diverse, urban, predominantly low-income city in northern California.

- Using a matched cohort design, we assessed whether the SLIV program was associated with increased student influenza vaccination, reduced community-wide influenza hospitalization, and reduced school absences.

- By the third and fourth years of the program, influenza vaccination coverage was 7%–11% higher among students in the SLIV site versus the comparison site. In those years, the SLIV program was associated with significantly lower influenza hospitalization rates among non-elementary-school-aged individuals and among the elderly. In addition, there were fewer absences due to illness in the SLIV site versus the comparison site in those years.

## What do these findings mean?

- A city-wide SLIV intervention was associated with increased influenza vaccination coverage, decreased illness-specific school absences among students, and lower influenza transmission community-wide, suggesting that the intervention may have produced herd effects.

## Introduction

Seasonal influenza contributes substantially to hospitalization and mortality, especially among infants and the elderly [1]. To prevent the spread of influenza, seasonal influenza vaccination of all individuals over 6 months of age has been recommended by the Advisory Committee on Immunization Practices (ACIP) in the US since 2010 [2]. The effectiveness of seasonal influenza vaccines varies from year to year depending on the quality of the influenza virus strain match and whether antigenic drift occurs between the time when the vaccine is manufactured and the start of the seasonal influenza epidemic, among other factors.

Because school-aged children are responsible for the greatest proportion of community-wide influenza transmission, efforts to increase vaccination among children are likely to have the largest impact on transmission [3–8]. Mathematical models estimate that vaccinating at least 50%–70% of school-aged children against influenza can prevent an influenza epidemic by producing herd immunity (i.e., "indirect effects") [9,10]. In recent years, influenza vaccination coverage in the US has ranged from 54% to 62% among elementary-school-aged children and from 37% to 44% among adults [11,12], lower than the Healthy People 2020 goal of 70% coverage [13]. Mathematical models project that influenza vaccination coverage of 80% in children would reduce influenza hospitalizations among children by 42% and among adults by approximately 20% [14].

School-located influenza vaccination (SLIV) programs have been proposed as a strategy to increase influenza vaccination coverage among children [15]. Prior studies reported that SLIV programs increased influenza vaccination [16–25] and decreased school absences [16–19,26–28] and student illnesses [16,19], and some studies report that the economic benefits of such programs likely outweigh the cost of program delivery [29,30]. There is some evidence that SLIV programs can produce community-wide indirect effects among preschool-aged children and adults; however, studies have produced conflicting results [20,21,31–33]. Many prior SLIV evaluations used study designs that are subject to confounding [17,18,20,21,26–28,31–33]; those that have used more rigorous designs did not measure health outcomes [23–25,34,35] or enrolled small numbers of schools [16,19]. To our knowledge, no prior studies have rigorously measured the impacts of large-scale SLIV interventions on student and community-wide health outcomes over multiple years.

Here, we report the findings of a 4-year evaluation of a SLIV program delivered to ≥95 preschools and elementary schools in Oakland, California, a diverse, urban, predominantly low-income city in northern California. The intervention was delivered city-wide to reduce influenza among elementary school children and to interrupt community-wide influenza transmission through herd effects. Using a matched cohort design and 3 independent data sources, we measured whether the intervention was associated with increased student influenza vaccination and decreased incidence of community-wide laboratory-confirmed influenza hospitalization and school absences.

## Methods

### Ethical statement

This study was approved by the Committee for Protection of Human Subjects at the University of California, Berkeley (Protocols 2014-01-5960 and 2016-12-9406). To measure influenza vaccination coverage, we invited caregivers of students to participate in a survey. Caregivers received a letter from the school district describing the purpose of the survey and providing details about the optional and anonymous nature of the survey. The UC Berkeley Committee for Protection of Human Subjects granted study investigators a waiver of documented informed consent to carry out the survey because in 2 years of pilot surveys in which we requested documented informed consent, the complexity of consent forms contributed to very low response rates that prevented us from collecting a sufficiently large sample to estimate vaccination coverage.

### SLIV intervention

The Shoo the Flu intervention (http://www.shootheflu.org) delivered free influenza vaccinations at all public and charter elementary schools in Oakland Unified School District (OUSD, the "intervention district") in the city of Oakland, California, and offered delivery to all other preschools and elementary schools in Oakland, including non-OUSD charter schools and private schools. Vaccinations were delivered prior to the start of influenza season from 2014 through 2018 (influenza seasons 2014–2015, 2015–2016, 2016–2017, and 2017–2018). OUSD enrolls a diverse, urban population of approximately 53,000 students, including over 26,000 elementary school students (kindergarten through grade 5). Over 70% of students in this district are from low-income households, and half of students speak a language other than English in their home. The intervention aimed to increase influenza vaccination coverage among primarily elementary-school-aged children in order to reduce influenza among elementary school children and to produce indirect effects protecting other age groups in the community. In its first 2 years, the intervention deployed a mass media campaign in the

Oakland area, including advertisements in the subway, bus shelters, billboards, and newspapers, as well as through digital media. The intervention did not carry out promotion efforts outside of the Oakland area, although it is possible that residents of areas near Oakland were exposed to Shoo the Flu media. Caregivers of the students provided written consent for vaccination, and this consent process was separate from consent to participate in the evaluation of the intervention. Children were eligible for vaccination regardless of their insurance status. From 2014 to 2018, between 95 and 138 elementary schools and preschools participated in Shoo the Flu, and each year the intervention vaccinated between 7,502 and 10,106 students (22%–28% of eligible students) (S1 Table).

### Influenza vaccine effectiveness during the intervention

In influenza seasons 2014–2015 and 2015–2016, the intervention offered the live attenuated influenza vaccine (LAIV) to students and the inactivated influenza vaccine (IIV) to students with LAIV contraindications, consistent with ACIP recommendations [36,37]. The intervention also offered IIV to school staff and teachers. In early 2016, the ACIP changed its recommendation for children aged 2 to 8 years from LAIV to IIV due to concerns about the low effectiveness of LAIV in the 2 prior seasons [38]. In influenza seasons 2016–2017 and 2017–2018, the intervention offered only IIV. The seasonal influenza vaccine delivered by the intervention had low effectiveness in 2014–2015 and 2015–2016 and moderate effectiveness in 2016–2017 and 2017–2018 (Fig 1) [39,40].

### Study design

This study is reported as per the Strengthening the Reporting of Observational Studies in Epidemiology (STROBE) guideline (S1 Checklist).

The SLIV intervention was offered to all preschools and elementary schools in the city of Oakland with the goal of delivering SLIV to the largest number of schools possible in order to interrupt influenza transmission (i.e., produce indirect effects) in the city. For this reason, it was infeasible to use a cluster-randomized design because all schools in the district were

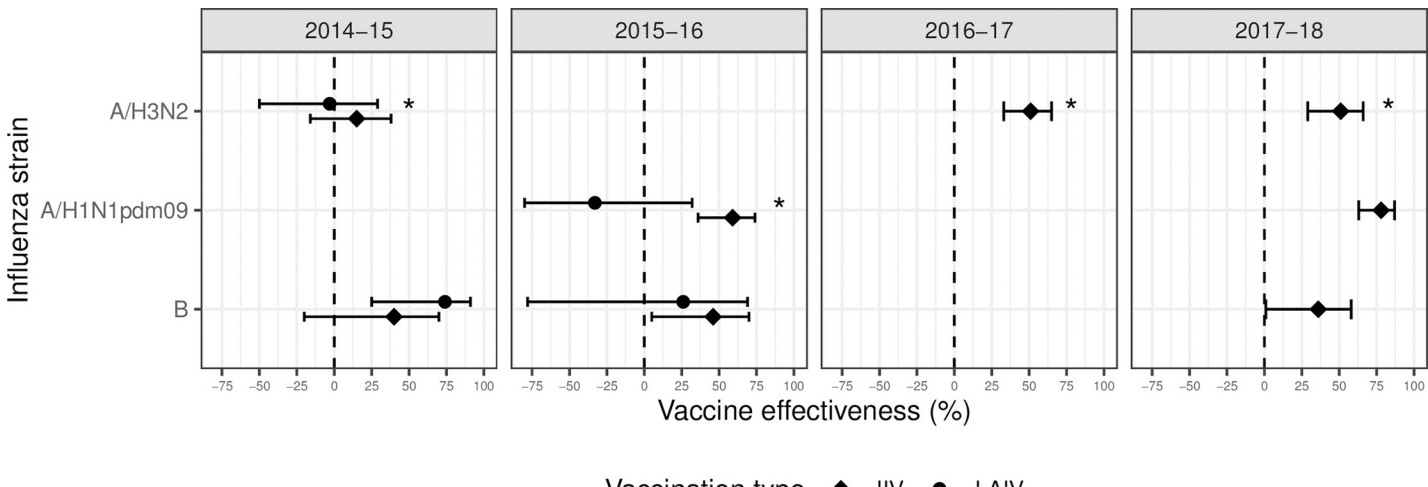

**Fig 1. Estimated influenza vaccine effectiveness from 2014 to 2018.** Vaccine effectiveness = $(1 - OR) \times 100\%$, where OR is the odds ratio for testing positive for influenza among individuals vaccinated for influenza compared to those who were not vaccinated. Estimates for influenza seasons 2014–2015 and 2015–2016 are for children 2–8 years; estimates for seasons 2016–2017 and 2017–2018 are for children 6 months to 8 years of age. Some strains for which there were not stable estimates for children 2–18 years are excluded from this plot. Source: US Influenza Vaccine Effectiveness Network. *Dominant strain circulating each season.

offered the intervention. The study used a matched cohort design to evaluate the SLIV intervention program using a retrospective analysis of prospectively collected data [41]. We drew on multiple independent data sources to assess a full range of outcomes that could have been affected by the SLIV intervention: (1) influenza vaccination coverage, (2) influenza hospitalizations in the community, and (3) all-cause and illness-specific school absence rates among elementary school students. We conducted a survey of a sample of student caregivers to measure influenza vaccination coverage and analyzed existing school absence and influenza hospitalization records. In some cases, outcome assessment required slightly different designs and estimators, as we describe below.

The matched design focused on the vaccination coverage survey. We first selected from comparison school districts among San Francisco Bay Area districts that had at least 4 elementary schools and had pre-intervention school-level characteristics similar to those of the intervention district. We restricted possible comparison districts to those with boundaries separated by at least 5 miles from the intervention district to minimize contamination. Though the intervention was provided to some private and charter schools in the intervention city, the study population was restricted to public elementary schools in the intervention city school district because pre-intervention data were not readily available to identify suitable comparison private or non-district charter schools.

We used a genetic multivariate matching algorithm [42] to pair-match schools in the intervention district with schools in each candidate comparison district. The matching algorithm used the following pre-intervention school-level characteristics: mean enrollment, class size, parental education, academic performance index scores, California standardized test scores, school-level percentage of English language learners, and school-level percentage of students receiving free lunch at school. We excluded preschools from this evaluation because the availability of preschools and enrollment criteria varied from school district to school district, complicating comparisons between districts. We identified West Contra Costa Unified School District (WCCUSD) as the best nearby comparison district because, on average, it had the smallest generalized Mahalanobis distance between paired schools [42] and a sufficient number of elementary schools ($N$ = 34 schools, grades K through 6) in the district to ensure adequate statistical power. The absolute value of the standardized difference was under 50 for most variables, indicating that the matching produced good quality school pair matches [43] (see S1 Appendix for further details).

Analyses of influenza hospitalization and school absences leveraged the matched design but used a slightly different approach tailored to each outcome. Because the intention was to measure community-wide indirect effects of SLIV on influenza hospitalization, we included hospitalizations of all residents of zip codes within the intervention and comparison district school catchment areas, including zip codes that were partially within the district boundary. To measure impacts on school absences, we prespecified inclusion of all public elementary schools enrolling kindergarten to grade 5 (K–5) (50 intervention schools, 34 comparison schools) rather than the matched subset used to design the vaccine coverage survey in order to maximize precision.

## Outcomes and data sources

**Vaccine coverage survey.** We conducted 2 cross-sectional surveys of student caregivers to measure caregiver-reported student influenza vaccination, including vaccine type and vaccine provider. A survey in March 2017 measured vaccination for influenza seasons 2014–2015, 2015–2016, and 2016–2017, and a survey in March 2018 measured vaccination for season 2017–2018. We distributed the surveys in 22 of the 34 matched school pairs (22 K–5 schools in

the intervention district and 22 K–6 schools in the comparison district). In all classrooms in each school, teachers distributed anonymous paper surveys to students to share with their caregivers. The survey was conducted independently from the intervention and allowed caregivers to report student influenza vaccination at any location.

**Laboratory-confirmed influenza hospitalizations.** We obtained counts of all laboratory-confirmed influenza hospitalizations, intensive care unit admissions, and deaths among hospitalized patients, and the duration of influenza hospitalizations from zip codes within the intervention and comparison school districts in influenza seasons 2011–2012 through 2017–2018 from the CDC-sponsored California Emerging Infections Program [44].

**School absence records.** We obtained records of absentee data for each student on each school day for school years 2011–2012 through 2017–2018 from all public elementary schools in each school district. Absences were classified by student grade, race/ethnicity, and absence type (all-cause absences versus illness-specific absences). Absences were classified as related to illness or not based on parent report, as recorded by each school district.

Statistical power calculations for each outcome are available in S2 Appendix.

## Statistical analysis

Unless otherwise specified, analyses were conducted in R version 3.6.1 [45]. Our pre-analysis plan, selected datasets, and replication scripts are available through the Open Science Framework (https://osf.io/c8xuq/). We defined overall effects as the difference in outcomes between elementary-school-aged individuals (both those who did and did not participate in the intervention) in the intervention versus comparison site, and indirect effects as the difference in outcomes between individuals in the intervention versus comparison site (Fig 2). We estimated indirect effects among non-elementary-school-aged individuals ($\leq$4 years or $\geq$13 years) and elderly individuals ($\geq$65 years).

## Definition of influenza seasons

Because influenza season timing varies from year to year, we prespecified a data-derived definition of influenza season in order to conduct our analysis during the weeks in which the

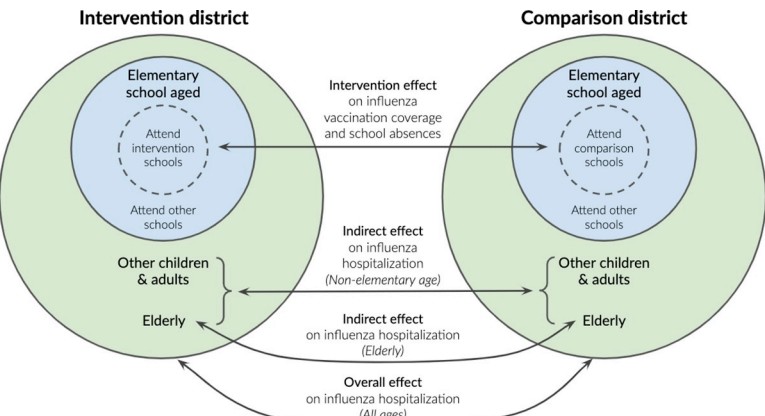

**Fig 2. Schematic of types of effects estimated in this study.** The "intervention effect" compares influenza vaccination coverage and school absence rates among elementary school children aged 5–12 years enrolled in schools participating in the intervention and schools in the comparison site. The "indirect effect" compares influenza hospitalization rates between individuals in zip codes overlapping with the intervention and comparison school districts; we estimated indirect effects among non-elementary-school-aged individuals (0–4 or $\geq$13 years) and elderly individuals ($\geq$65 years). The "overall effect" compares influenza hospitalization rates among all individuals in zip codes overlapping with the intervention and comparison school districts; it averages across the intervention effect and indirect effect.

influenza epidemic occurred locally each year. Under this definition, influenza season started when there were at least 2 consecutive weeks in which the percentage of medical visits for influenza-like illness in California as reported by the California Department of Public Health [46] exceeded a cutoff, and the season ended when there were at least 2 consecutive weeks in which the percentage was less than or equal to the cutoff. In our pre-analysis plan, we set the cutoff at 2%. Post hoc, we examined seasons defined using cutoffs of 2%, 2.5%, and 3% and selected 2.5% as the primary definition because, in some seasons, the 2% cutoff included weeks as early as September and as late as June, and we felt that the 2.5% cutoff best captured seasonal variation in peak influenza-like illness (S1 Fig). We made this change only considering overall seasonal patterns for influenza-like illness before estimating any program effects.

**Influenza vaccination coverage.** We estimated influenza vaccination coverage and 95% confidence intervals using robust sandwich standard errors that accounted for clustering at the school level [47]. We estimated differences in vaccination coverage between districts using a generalized linear model that adjusted for student race/ethnicity and caregiver's education level and estimated standard errors that accounted for clustering within matched school pairs. We restricted the analysis to grades K–5 because the intervention district's elementary schools did not include sixth grade. To assess possible selection bias among the sample of caregivers who responded to the survey, we also estimated vaccination coverage after standardizing the distributions of race/ethnicity and education in the sample to the pre-intervention percentages using data from the California Department of Education for the 44 participating schools and for the entire districts (see details in S3 Appendix).

**Laboratory-confirmed influenza hospitalization.** To estimate the cumulative incidence of influenza hospitalization, we obtained age- and race-specific population counts from the 2010 US Census in the same set of zip codes used to identify influenza cases. We also obtained more recent annual population counts from the 5-year American Community Survey (ACS). Because counts were similar between both data sources and the ACS did not provide population counts by race and age in years, we used the US Census data in our primary analyses. We fit log-linear Poisson models to estimate cumulative incidence using a log population offset and adjusting for age, sex, and race [48]. To control for pre-season differences between districts, we estimated the difference-in-differences (DID), defined as the difference in incidence in the intervention district prior to (2011–2013) and during the intervention minus the difference in incidence in the comparison district prior to and during the intervention. Examination of pre-intervention influenza hospitalization patterns indicated that the equal trends assumption was reasonable (Fig 3; S4 Appendix Figs A and B). The DID parameter eliminates any time-invariant confounding and accounts for differences in pre-intervention outcomes between districts [49]. We obtained standard errors for each quantity using the delta method. Consistent with our pre-analysis plan, we did not estimate DID for intensive care unit admission on its own or for influenza mortality because these are rare outcomes, and we were likely to be underpowered to detect an effect. We also estimated outcomes in the peak week of the influenza season, defined as the week with the highest rate of influenza hospitalization in the study site. We performed a sensitivity analysis using the following alternative influenza season cutoffs: (1) 2 consecutive weeks in which the percentage of medical visits for influenza-like illness exceeded (season start) or fell below (season end) 2% or 3% (instead of 2.5%) and (2) October 1 (season start) to April 30 (season end) each year. The latter classification is conservative in that it often includes many weeks of the year with limited influenza-like illness in the influenza season.

**School absence rates.** We restricted the primary analysis of school absence rates to school days when both districts were in session. In addition, we restricted to school days that occurred during influenza season because we did not expect the intervention to influence influenza and

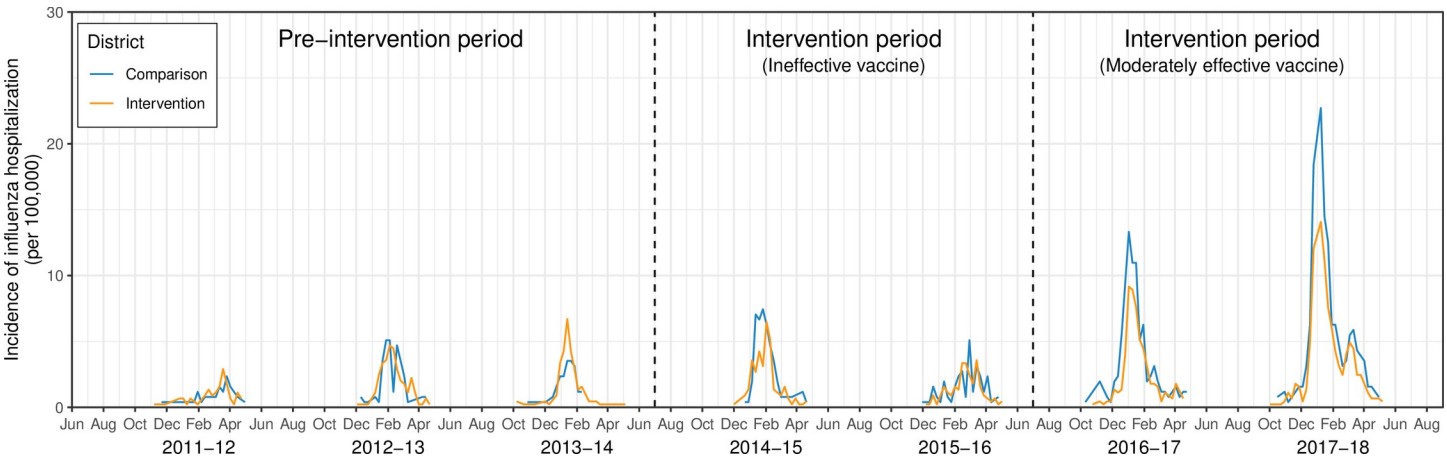

**Fig 3. Weekly incidence of inpatient laboratory-confirmed influenza prior to and during the intervention among all ages.** Weekly incidence proportion of laboratory-confirmed influenza hospitalizations (including intensive care unit admissions) between October 1 and April 30 of each year. Hospitalizations included school district residents tested at healthcare facility laboratories located in zip codes overlapping with OUSD and WCCUSD (Alameda County Public Health Department, Children's Hospital Oakland, Contra Costa Public Health Department, Kaiser Permanente, Sutter Health). Population denominators were obtained from the 2010 US Census using the same set of zip codes.

absenteeism outside of that period. We also estimated outcomes in the peak week of the influenza season, defined as the week with the highest proportion of influenza-like illness visits in California when school was in session. Examination of pre-intervention school absence patterns indicated that the equal trends assumption required for valid DID analysis was reasonable (S4 Appendix Figs C and D). We estimated DID in mean absence rates using linear regression models and adjusted for available time-varying covariates: student race, grade, and month of absence. We did not adjust for pre-intervention school characteristics (i.e., those used in the matching of school pairs) because they were time-invariant and thus would have no effect on DID estimates. We calculated 95% confidence intervals using robust standard errors that accounted for clustering within schools [47]. We estimated the difference in total student absences during influenza season by multiplying DID estimates and confidence interval bounds by the total student enrollment and the number of school days in each influenza season.

Differential measurement error could have occurred if absences were recorded with different levels of accuracy between the intervention and comparison site. To detect potential differential measurement error of school absences, we conducted a negative control analysis [50] using the school days in August, September, May, and June; these months were prior to the delivery of vaccines at school and outside of influenza season, when we did not expect to see an effect of the intervention [14]. We also performed a sensitivity analysis using the following alternative influenza season cutoffs: (1) 2 consecutive weeks in which the percentage of medical visits for influenza-like illness exceeded (season start) or fell below (season end) 2% or 3% and (2) week 40 of each year to week 20 of the next year. The latter classification is used in influenza surveillance by the CDC, but it often includes periods with limited influenza-like illness.

As we describe in the Results, the negative control analysis indicated possible differential measurement error of school absences, so we performed a post hoc probabilistic bias analysis to quantify the possible influence of outcome misclassification on our results [51]. In this analysis, we assumed a range of possible values of the sensitivity and specificity of absence classification based on conversations with data managers in each school district (S5 Appendix

Table A and Fig A). For example, in the comparison district, we assumed that the sensitivity of all-cause absences (the probability that a true absence of any cause was classified as such by the school district) was most likely to be close to 1 and in very few instances was less than 0.50. In a simulation, we drew from the distributions of sensitivity and specificity to calculate the bias-corrected absence rate correcting for possible misclassification under a range of plausible scenarios; we repeated this process 1,000 times to obtain distributions of bias-corrected DID estimates (S5 Appendix). We focused on outcome misclassification because exposure misclassification was highly unlikely, and our DID analysis accounted for measured and time-invariant unmeasured confounders.

To examine impact across different levels of program participation among the 50 SLIV intervention schools, we predicted the mean absence rate ($Y$) setting each school's value to each observed level of school participation in SLIV ($A$) and adjusting for the school-level covariates student race/ethnicity, the percentage of students in each grade, average enrollment, mean class size, mean parent education level, percentage of English language learners, percentage of students receiving free lunch, mean 2012 Academic Performance Index score, mean 2013 Academic Performance Index score, and mean California Standards Test scores ($W$) ($\sum_w E[Y|A = a, W = w]P(W = w)$). We used an ensemble machine learning algorithm that flexibly adjusted for covariates correlated with the outcome ($p < 0.1$) in statistical models. The algorithm included the following estimation methods: the simple mean, main effects generalized linear models, stepwise logistic regression, Bayesian generalized linear models [52], generalized additive models [53], elastic net regression [54], random forest [55], and gradient boosting [56].

## Results

### Influenza vaccine coverage

Many measured pre-intervention characteristics were similar in populations residing in the catchment areas of the intervention and comparison schools (Table 1). The median household income was lower in the intervention site ($51,849, 95% CI $50,460, $53,238) than the comparison site ($61,596, 95% CI $61,596, $63,530). Comparing the intervention site to the comparison site, the proportion black/African American residents was higher (26%, 95% CI 25%, 27%, versus 17% 95% CI 16%, 18%), and the percentage of Hispanic or Latino residents was lower (26%, 95% CI 25%, 27%, versus 33%, 95% CI 32%, 35%). The percentage of residents with a bachelor's degree or higher was 15% (95% 12%, 17%) in the intervention site and 6% (95% CI 4%, 8%) in the comparison site.

In March 2017, field staff disseminated 8,121 surveys in 22 schools in OUSD and 10,054 surveys in 22 schools in WCCUSD (S2 Fig). The response rates were 28% ($N = 2,246$ surveys) in OUSD and 38% in WCCUSD ($N = 3,824$). One school in OUSD withdrew, and we excluded its matched pair from analyses comparing the 2 districts. In March 2018, the same schools were invited to participate, and the response rates were similar. In each survey, 32%–40% of respondents had a higher than high school level of education; the education level among survey respondents was lower than in the school district catchment areas as a whole (Table 1), which included households that participated in this survey as well as those whose children attended private schools. Approximately 22%–27% of respondents' primarily language spoken at home was Spanish. The most common student race/ethnicity was Latino (36%–41% in intervention; 50%–51% in comparison), followed by Asian (22%–25% in intervention; 16%–17% in comparison) and black/African American (16% in intervention; 10% in comparison).

Influenza vaccination coverage (from any source) among K–5 elementary students did not differ statistically between the intervention and comparison districts in the first 2 years of the

**Table 1. Pre-intervention characteristics of the population in the school district catchment areas.**

| Characteristic | Intervention (95% CI) | Comparison (95% CI) |
|---|---|---|
| Median household income (dollars) | 51,849 (50,460, 53,238) | 61,596 (59,662, 63,530) |
| Households below the poverty level (%) | 21 (20, 22) | 15 (13, 16) |
| Highest education level (%) | | |
| Less than high school | 16 (15, 18) | 14 (12, 17) |
| High school graduate | 24 (21, 26) | 30 (25, 34) |
| Some college or associate's degree | 46 (43, 48) | 50 (46, 55) |
| Bachelor's degree or higher | 15 (12, 17) | 6 (4, 8) |
| Children attending kindergarten in private versus public schools (%) | | |
| Public | 87 (81, 92) | 86 (80, 92) |
| Private | 13 (8, 19) | 14 (8, 20) |
| Children attending grade 1–4 in private versus public schools (%) | | |
| Public | 89 (86, 92) | 84 (79, 88) |
| Private | 11 (8, 14) | 16 (12, 21) |
| Children attending grade 5–8 in private versus public schools (%) | | |
| Public | 89 (86, 91) | 87 (83, 91) |
| Private | 11 (9, 14) | 13 (9, 17) |
| Race (%) | | |
| White | 41 (40, 42) | 48 (47, 50) |
| Black or African American | 26 (25, 27) | 17 (16, 18) |
| Asian | 16 (16, 17) | 19 (18, 20) |
| Other race | 9 (8, 10) | 8 (7, 9) |
| Native Hawaiian and other Pacific Islander | 1 (0, 1) | 0 (0, 1) |
| 2 or more races | 6 (6, 7) | 6 (5, 7) |
| Hispanic or Latino ethnicity (%) | 26 (25, 27) | 33 (32, 35) |

Data from the 3-year 2013 American Community Survey subset by school district boundaries.

SLIV intervention but was higher in the intervention district in the latter 2 years of the intervention. In relation to the comparison district, influenza vaccination coverage in the intervention district was 7% (95% CI 4%, 11%; $p < 0.001$) higher in influenza season 2016–2017 and 11% (95% CI 7%, 15%; $p < 0.001$) higher in season 2017–2018; differences were statistically significant (Table 2). Standardizing vaccination coverage by student race and parent education

**Table 2. Caregiver-reported influenza vaccination coverage among elementary school students during the school-located influenza vaccination intervention period.**

| Season | Intervention | | Comparison | | Intervention minus comparison[a], percent (95% CI) | p-Value |
|---|---|---|---|---|---|---|
| | N | Percent (95% CI) | N | Percent (95% CI) | | |
| 2014–2015[b] | 2,246 | 59 (56, 63) | 3,824 | 64 (61, 67) | −5 (−9, −1) | 0.017 |
| 2015–2016[b] | 2,246 | 68 (65, 70) | 3,824 | 66 (64, 68) | 1 (−2, 4) | 0.416 |
| 2016–2017[b] | 2,246 | 64 (61, 67) | 3,824 | 56 (54, 59) | 7 (4, 11) | <0.001 |
| 2017–2018[c] | 2,421 | 64 (60, 69) | 4,086 | 53 (51, 56) | 11 (7, 15) | <0.001 |

[a]Difference in percentage vaccinated adjusting for student race and caregiver education level. All confidence intervals were calculated using robust standard errors accounting for clustering at the school level.

[b]Influenza vaccination was reported by caregivers in March 2017.

[c]Influenza vaccination was reported by caregivers in March 2018.

produced similar results (S3 Fig). The percentage of elementary students vaccinated for influenza at school was 14% in 2014–2015, 23% in 2015–2016, 24% in 2016–2017, and 26% in 2017–2018 (S4 Fig; S2 Table). The majority of students not vaccinated at school were vaccinated at a doctor's office or health clinic. In 2014–2015 and 2015–2016, seasons when IIV and LAIV were available, 48%–52% of students received IIV and 12% of students received LAIV in the comparison site, and 36%–39% of students received IIV and 19%–23% of students received LAIV in the intervention site (S5 Fig; S3 Table). When adjusting for school-level characteristics, influenza vaccination coverage did not vary by the percentage of students participating in SLIV in each school (S6 Fig).

## Influenza hospitalization

In the 3 influenza seasons before the intervention, the age-standardized incidence of influenza-related hospitalization was similar between the intervention and comparison districts (Fig 3). In 2014–2015 and 2015–2016, the incidence of influenza hospitalization was not statistically different between the intervention and comparison districts in any age group (Fig 3; S4 Table). In 2016–2017 and 2017–2018, hospitalization incidence was lower in the intervention versus comparison district in all age groups. Among non-elementary-aged individuals (0–4 or ≥13 years), the DID in the cumulative incidence of influenza hospitalization was −17 (95% CI −30, −4; $p = 0.008$) in 2016–2017 and −37 (95% CI −54, −19; $p < 0.001$) in 2017–2018 (Fig 3; S4 Table) in the intervention versus comparison district. Among individuals aged at least 65 years, the DID in the cumulative incidence of influenza hospitalization was −73 (95% CI −147, 1; $p = 0.054$) in 2016–2017 and −160 (95% CI −267, −53; $p = 0.004$) in 2017–2018 (Fig 4; S4

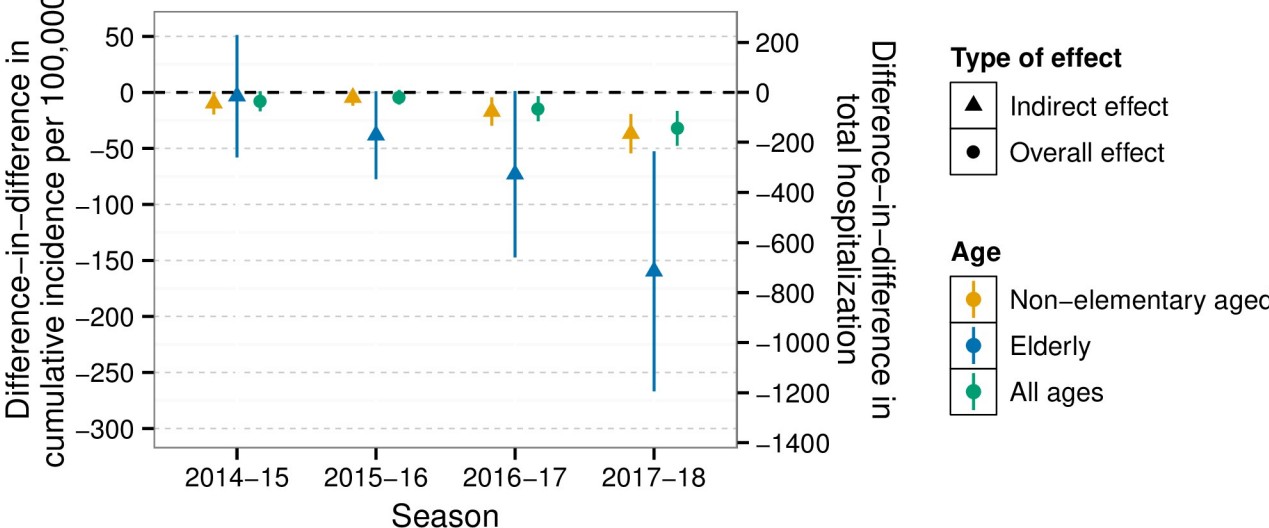

**Fig 4. Total and indirect effects on cumulative incidence of inpatient laboratory-confirmed influenza during influenza season.** Cumulative incidence of laboratory-confirmed influenza hospitalization (including intensive care unit admissions) during influenza season. Difference-in-differences estimates represent the difference between intervention and control groups in their change in incidence from the 3 pre-program influenza seasons (2011–2013) to each program season, which removes any time-invariant differences between groups (measured or unmeasured). The left $y$-axis presents the difference-in-differences in the cumulative incidence per 100,000. The right $y$-axis presents the difference-in-differences in the total hospitalizations, which was calculated as the product of the difference-in-differences in the cumulative incidence and the population of the intervention site. For both $y$-axes, triangles represent indirect effects and circles represent overall effects. Parameters were estimated using a log-linear Poisson model with an offset for population size, and were further adjusted for age, race, and sex. Standard errors and 95% confidence intervals were obtained using the delta method. We defined influenza season based the percentage of medical visits for influenza-like illness in California as reported by the California Department of Public Health. Influenza season started when there were at least 2 consecutive weeks in which the percentage of medical visits for influenza-like illness exceeded 2.5%, and the season ended when there were at least 2 consecutive weeks in which the percentage was less than or equal to 2.5%.

Table). Results were similar in analyses restricted to the peak week of influenza hospitalization (S7 Fig).

Among all ages and across all 4 influenza seasons, the mean DID in length of influenza hospitalization was approximately 1 to 2.5 days lower in the intervention district versus the comparison district (S8 and S9 Figs). The incidence of influenza-related intensive care unit admissions was lower in the intervention site than in the comparison site in seasons 2014–2015, 2015–2016, and 2016–2017 and was similar in season 2017–2018 (S10 Fig). The influenza-related mortality rate was slightly higher in the intervention site before the intervention and was lower in the 2014–2015, 2015–2016, and 2017–2018 influenza seasons (S11 Fig). However, for all intensive care unit and mortality estimates, 95% confidence intervals for district-specific estimates overlapped. Our sensitivity analyses using alternative population denominators, influenza case definitions, and influenza season definitions yielded similar results overall (S12 and S13 Figs).

## School absenteeism

In the 3 years prior to the Shoo the Flu program, during influenza season, the mean absence rate per 100 days in the intervention versus comparison district was 4.85 versus 5.84 for all-cause absences and 2.84 versus 2.81 for illness-specific absences (S5 Table). The DID in mean illness-specific absence rate per 100 days was −0.16 (95% CI −0.54, 0.23; $p = 0.425$) in 2014–2015 and −0.34 (95% CI −0.78, 0.10; $p = 0.130$) in 2015–2016 (Fig 5; S5 Table). In 2016–2017 and 2017–2018, the DID in illness-specific absence rate per 100 days was lower in the intervention district compared to the comparison district (2016–2017 DID −0.63 [95% CI −1.14, −0.13; $p = 0.014$]; 2017–2018 DID −0.80 [95% CI −1.28, −0.31; $p = 0.001$]). The reduction in total illness-specific student absences during influenza season was 3,538 (95% CI 709, 6,366; $p = 0.014$) in 2016–2017 and 8,249 (95% CI 3,213, 13,285; $p = 0.001$) in 2017–2018 in the intervention district. For all-cause absences, the DID estimates during influenza season were not statistically significant in any years of the program. During the peak week of the influenza season, there was evidence of larger reductions in illness-specific absence rates in 2014–2015, 2016–2017, and 2017–2018, and there was a significant reduction in all-cause absences in 2017–2018 (S5 Table). Our sensitivity analyses using alternative influenza season definitions were consistent with the primary analysis (S14 Fig). Mean absence rates were not associated with the percentage of students in each school that participated in Shoo the Flu in 2014–2015 and 2015–2016. In 2017–2018, the school-level SLIV participation rate was associated with a modest reduction in the mean absence rate when adjusting for potential school-level confounders (S15 Fig).

We performed a negative control time period analysis, estimating DID outside influenza season, when we did not expect the intervention to affect absence rates. Overall, we did not see an effect on all-cause absences outside of influenza season. However, there were statistically significant reductions in illness-specific absences outside of influenza season in influenza seasons 2015–2016, 2016–2017, and 2017–2018 (S16 Fig), suggesting that differential measurement error may have impacted the primary analysis. We explored the influence of outcome misclassification on our findings with a probabilistic bias analysis under assumed distributions of sensitivity and specificity of outcome classification. We found that the majority of bias-corrected DID estimates in 2016–2017 and 2017–2018 indicated a reduction in both types of absences in the intervention district (S5 Appendix Figs B and C). These findings suggest that outcome misclassification was not strong enough to alter the scientific inferences in our primary analysis of absenteeism.

## Discussion

Here, we evaluated the impact of a city-wide SLIV intervention delivered to 95 or more elementary schools per season in a diverse, predominantly low-income city. During the first 2

## A) Difference−in−differences in mean absences

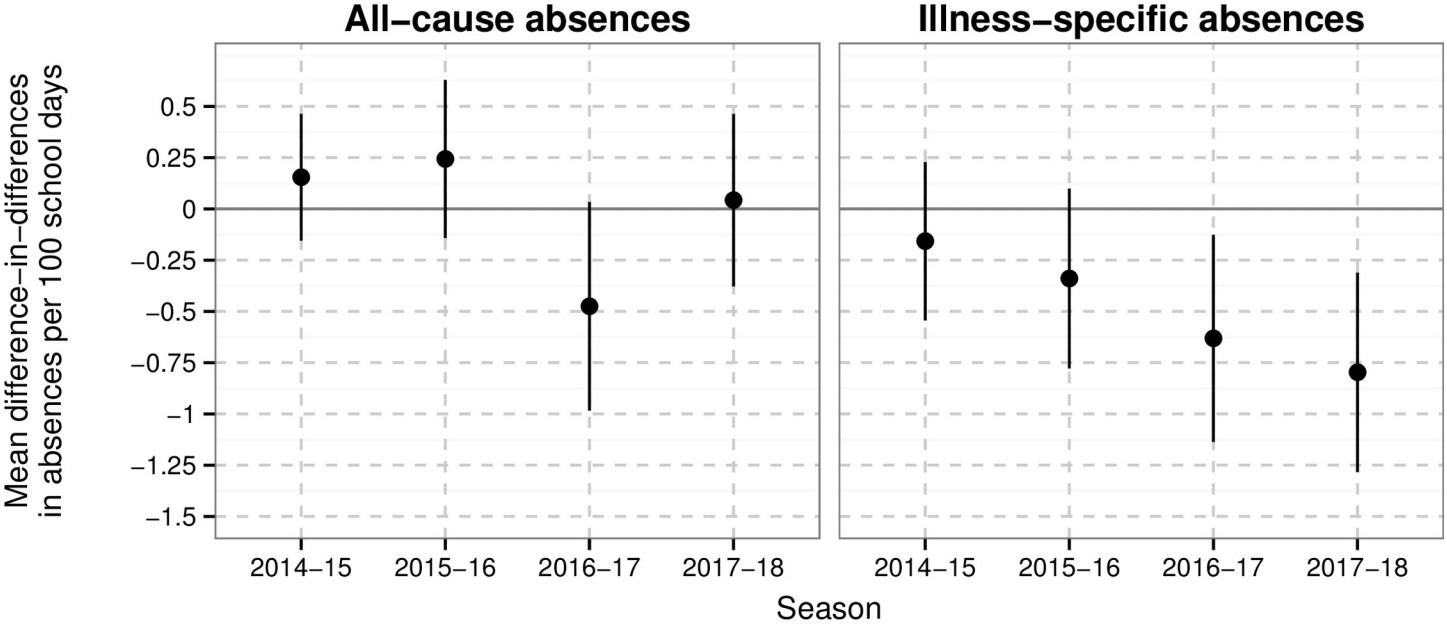

## B) Difference−in−differences in total absences

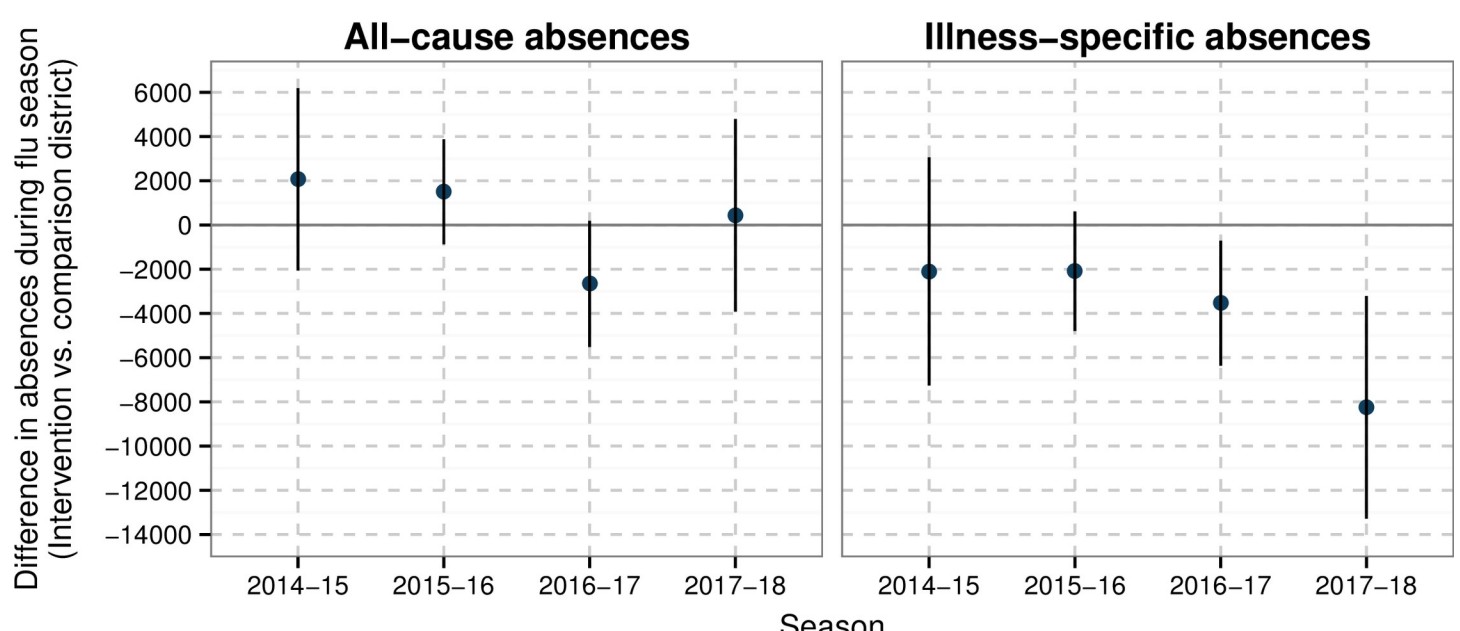

**Fig 5. Intervention effects on the school absence rate per 100 school days during influenza season.** In (A), each difference-in-differences estimate compares the difference in mean absence rate in each district in a program influenza season compared to the 3 pre-program seasons (2011–2013); in (B), each difference-in-differences estimate compares the difference in total absence rate, which was calculated by multiplying difference-in-differences in mean absences by the total enrollment and total number of school days during influenza season each season. Difference-in-differences parameters remove any time-invariant differences between groups (measured or unmeasured). Parameters were estimated using a generalized linear model and were adjusted for month, student race, and grade. Standard errors and 95% confidence intervals account for clustering at the school level. We defined influenza season based the percentage of medical visits for influenza-like illness in California as reported by the California Department of Public Health. Influenza season started when there were at least 2 consecutive weeks in which the percentage of medical visits for influenza-like illness exceeded 2.5%, and the season ended when there were at least 2 consecutive weeks in which the percentage was less than or equal to 2.5%. Note: in 2011–2012, 2016–2017, and 2017–2018, the peak week of the percentage of influenza-like illness visits in California was the last week of December, which coincided with school

breaks, so for the absentee analysis we shifted the peak week definition to the week before or after the school break (when both school districts were in session) that had the higher percentage of influenza-like illness visits.

years of SLIV, the program offered LAIV, which had low effectiveness [57,58], and on the whole, we did not observe associations between SLIV and hospitalization or absence rates in those years. In the 2016–2017 and 2017–2018 influenza seasons, when the intervention delivered IIV and the vaccine was moderately effective [40,59], we observed an association between SLIV and reduced illness-related school absences and reduced hospitalizations among age groups not targeted by SLIV, suggesting that the intervention produced indirect effects.

Unique strengths of our study design include the use of multivariate matching and a DID approach to minimize systematic differences between the intervention and comparison sites and the prespecification of our statistical analysis plan [60]. Evaluating SLIV over multiple years enabled us to examine the impact of SLIV in influenza seasons with different levels of vaccine effectiveness, vaccine recommendations, and circulating strains of influenza. Furthermore, this study leveraged 3 distinct, independent data sources that provided internally consistent results.

The SLIV intervention was associated with increases in influenza vaccination coverage of up to 11 percentage points among elementary school students in the intervention site versus the comparison site. This increase in vaccination coverage is smaller than those reported in prior SLIV studies, which ranged from 7 to 41 percentage points. However, in most prior studies, coverage at baseline or in the comparison group was substantially lower than 50% [16,17,19,21,23–25,33], while in our study it was 53%–66%. It may be more difficult for SLIV to substantially increase coverage at moderate pre-intervention coverage levels. In addition, the switch from LAIV to IIV in the third year of the Shoo the Flu intervention may have inhibited larger increases in coverage because many children and/or caregivers prefer the nasal spray generally and in a school setting, and media coverage of poor LAIV effectiveness may have increased vaccine hesitancy for all influenza vaccine formulations. In the comparison site, the 10% reduction in vaccination coverage between 2015–2016 and 2016–2017, when LAIV was discontinued, was on par with the approximately 12% of students reported by caregivers to have been vaccinated with LAIV in 2014–2015 and 2015–2016 in that site (S5 Fig; S3 Table). Yet, vaccination coverage in the intervention district grew relative to the comparison district over time, in part because the comparison district coverage did not recover from the decline associated with the switch from LAIV to IIV vaccines. It is possible that coverage in the intervention district will continue to grow in future years as the intervention builds trust and recognition.

A key question is whether SLIV interventions increase vaccination coverage among students who would otherwise not be vaccinated, whether they merely shift vaccination location from healthcare providers to schools, or whether both occur. Interventions that vaccinate children who would otherwise not be vaccinated will have the largest impact on influenza transmission. Our findings suggest that both phenomena may have occurred in this study. Coverage was higher in the intervention versus comparison district in the final 2 years of the evaluation, suggesting that children who otherwise would not have been vaccinated were vaccinated by the SLIV intervention. In addition, the proportion of students vaccinated at school in the intervention district (24%–26% in the latter 2 years [S4 Fig]) exceeded the difference in coverage between districts (7%–11%). While we cannot definitively determine what the vaccination coverage levels would have been in the intervention site in the absence of the SLIV intervention, these findings suggest that approximately 15% of students vaccinated by the SLIV intervention may otherwise have been vaccinated through other means. A limitation is

that pre-intervention vaccination data were not available, so we were not able to conduct DID analyses to account for any pre-SLIV differences in vaccination coverage between districts.

Influenza vaccination coverage in the Shoo the Flu intervention site was 64%, which was up to 11% higher than in the comparison site and well within the 50%–70% range in which herd immunity is expected [9,10]. We observed reductions in influenza hospitalization among non-elementary-school-aged community members in seasons with moderate vaccine effectiveness (S4 Table). Indirect effects were strongest among individuals aged 65 years or older—the age group most vulnerable to influenza hospitalization and mortality. The magnitude of indirect effects we observed is similar to those of other SLIV interventions [20,21,31–33] and of interventions vaccinating children at any location [61]. In addition, our results are highly consistent with mathematical models, which project that an increase from 40% to 60% coverage in children aged 6 months to 18 years would reduce influenza hospitalization among adults 19–64 years of age by 36% and adults 65 years or older by 33% [14]. Our findings suggest that even modest increases in vaccination (i.e., up to 11%) associated with SLIV can produce meaningful community-wide reductions in influenza hospitalization, consistent with mathematical models [62].

This analysis is subject to several limitations. First, although the reductions in illness-specific school absences we observed were of a similar magnitude to those reported in prior studies [16–19,26–28], our finding of significant differences in absence rates outside of influenza season suggests that our absentee results may be subject to differential misclassification. It is possible that school-year- and school-district-specific differences unrelated to the SLIV intervention could explain these findings. For example, district-specific policies to decrease absences around school breaks (some of which coincide with influenza season) or at the beginning or the end of the school year (which is included in our negative control time period analysis) may have impacted our estimates in an unknown direction, especially if they differed before and during the SLIV intervention. In addition, parents may have attributed illness absences to a different reason, and such misclassification could have varied by district. Our probabilistic bias correction analysis suggested that correcting for misclassification would not change our conclusion that the SLIV intervention reduced illness-specific absences in 2016–2017 and 2017–2018.

Second, given that the SLIV intervention was delivered city-wide, it was not possible to conduct a randomized trial. The matched cohort design minimized differences in measured confounders between the intervention and comparison site, and DID analyses controlled for measured and unmeasured time-invariant confounding. Nevertheless, unmeasured time-dependent confounding could still bias this observational design. For example, the first year of the SLIV intervention coincided with the rollout of the preventive benefits of the Affordable Care Act, which may have jointly affected healthcare utilization and vaccination patterns in the study region.

Third, our vaccination coverage estimates relied on caregiver reporting, which is subject to inaccurate recall and low response rates. Prior studies report that caregiver recall of child influenza vaccination in the past season has a sensitivity of 88%–92% and a specificity of 82%–90% compared to medical records [63–65]. Coverage estimates for 2014–2015 and 2015–2016 may be more vulnerable to measurement error because they rely on a 2- to 3-year recall period. Overall, our vaccination coverage estimates were consistent with caregiver-reported national and California-specific estimates from the Centers for Disease Control and Prevention [11,66]. Nevertheless, it is possible that caregiver recall in the intervention district differed from that in the comparison district; the presence of the SLIV intervention may have increased parents' likelihood of accurately recalling their child's vaccination status. It was not possible to validly compare our results to coverage estimates from the California Immunization Registry because

prior to and during the SLIV intervention, many local vaccine providers did not consistently enter data into the registry as there is no mandate to do so in California, and there were differences in reporting rates between providers in the intervention and comparison site.

Finally, we did not have a direct measure of laboratory-confirmed influenza incidence in elementary school children or influenza vaccination coverage estimates among non-elementary-aged individuals; thus, it remains possible that factors other than the SLIV intervention could explain our findings. We were not able to link individuals between different data sources because personal identifiers were not available to us. Nevertheless, high levels of internal consistency across results from 3 independent data sources lend credence to the validity of our findings.

## Conclusions

Offering SLIV to all elementary schools in a large, urban district was associated with 7%–11% increases in vaccination, which were followed by meaningful reductions in illness-specific absences among school children and community-wide influenza hospitalization among those not targeted by the program, including the elderly, in 2016–2017 and 2017–2018. Our findings suggest that even modest increases in influenza vaccination above moderate coverage levels are associated with broad community-wide benefits.

## Supporting information

**S1 Checklist. STROBE checklist.**
(DOCX)

**S1 Appendix. Selection of schools for the vaccine coverage survey.**
(PDF)

**S2 Appendix. Statistical power calculations.**
(PDF)

**S3 Appendix. Assessment of selection bias in the vaccine coverage survey.**
(PDF)

**S4 Appendix. Pre-intervention influenza hospitalization and school absence rates in each site.**
(PDF)

**S5 Appendix. Quantitative bias analysis to assess misclassification of absence rates.**
(PDF)

**S1 Fig. Alternative influenza season definitions.**
(PDF)

**S2 Fig. Vaccine coverage survey participant flow.**
(PDF)

**S3 Fig. Standardized percent of students vaccinated for influenza from all sources among elementary school students in 44 OUSD and WCCUSD schools for 2014–2018.**
(PDF)

**S4 Fig. Percentage of elementary students vaccinated for influenza by vaccination location in each district.**
(PDF)

**S5 Fig. Percentage of elementary students whose caregiver reported they were vaccinated for influenza by vaccine type in each district.**
(PDF)

**S6 Fig. Relationship between caregiver-reported influenza vaccination coverage among elementary students and school-level participation in the SLIV program in intervention schools.**
(PDF)

**S7 Fig. Overall and indirect effects on cumulative incidence of inpatient laboratory-confirmed influenza during the peak week of influenza hospitalization.**
(PDF)

**S8 Fig. Overall and indirect effects on length of influenza hospitalization excluding outlier.**
(PDF)

**S9 Fig. Overall and indirect effects on length of influenza hospitalization including outlier.**
(PDF)

**S10 Fig. Cumulative incidence of influenza-related intensive care unit (ICU) admission per 100,000 by influenza season and site.**
(PDF)

**S11 Fig. Cumulative incidence of influenza mortality per 100,000 by influenza season and site.**
(PDF)

**S12 Fig. Sensitivity analyses estimating overall and indirect effects on cumulative incidence of inpatient laboratory-confirmed influenza with alternative numerators and denominators.**
(PDF)

**S13 Fig. Sensitivity analyses estimating overall and indirect effects on cumulative incidence of inpatient laboratory-confirmed influenza using alternative influenza season definitions.**
(PDF)

**S14 Fig. Sensitivity analyses estimating difference-in-differences in school absence rates using alternative influenza season definitions.**
(PDF)

**S15 Fig. Relationship between absence rates and school-level participation in the SLIV program in intervention schools.**
(PDF)

**S16 Fig. Intervention effects on the school absence rate per 100 school days stratifying by month and influenza season.**
(PDF)

**S1 Table. Influenza vaccines delivered by the SLIV intervention each year.**
(PDF)

**S2 Table. Percentage of elementary students vaccinated for influenza by vaccination location in each district.**
(PDF)

**S3 Table. Percentage of elementary students whose caregiver reported they were vaccinated for influenza by vaccine type in each district.**
(PDF)

**S4 Table. Difference-in-differences in cumulative incidence and total number of laboratory-confirmed influenza hospitalization per 100,000 during influenza season.**
(PDF)

**S5 Table. School absence rate per 100 days and difference-in-differences in absence rate during influenza season.**
(PDF)

## Acknowledgments

We would like to thank Catharine Ratto for technical advising as well as our partners at Shoo the Flu, the Alameda County Public Health Department, OUSD, WCCUSD, the California Emerging Infections Program, and Applied Survey Research for their contributions to this evaluation.

## Author Contributions

**Conceptualization:** Jade Benjamin-Chung, Benjamin F. Arnold, Alan E. Hubbard, Arthur Reingold, John M. Colford, Jr.

**Data curation:** Jade Benjamin-Chung, Kunal Mishra, Nolan Pokpongkiat, Anna Nguyen, Pam D. Kirley, Tanya Libby.

**Formal analysis:** Jade Benjamin-Chung, Chris J. Kennedy, Kunal Mishra, Alan E. Hubbard.

**Funding acquisition:** Jade Benjamin-Chung, Benjamin F. Arnold, Alan E. Hubbard, Arthur Reingold, John M. Colford, Jr.

**Investigation:** Jade Benjamin-Chung, Kate Holbrook, Erica Pan, Pam D. Kirley, Tanya Libby.

**Methodology:** Jade Benjamin-Chung, Benjamin F. Arnold, Chris J. Kennedy, Alan E. Hubbard, Arthur Reingold, John M. Colford, Jr.

**Project administration:** Jade Benjamin-Chung.

**Resources:** Kate Holbrook, Erica Pan, Pam D. Kirley, Tanya Libby.

**Software:** Jade Benjamin-Chung, Chris J. Kennedy, Kunal Mishra, Nolan Pokpongkiat, Anna Nguyen, Wendy Jilek, Alan E. Hubbard.

**Supervision:** Jade Benjamin-Chung, Benjamin F. Arnold, Alan E. Hubbard, Arthur Reingold, John M. Colford, Jr.

**Validation:** Jade Benjamin-Chung, Chris J. Kennedy, Kunal Mishra, Nolan Pokpongkiat, Anna Nguyen.

**Visualization:** Jade Benjamin-Chung, Chris J. Kennedy, Kunal Mishra, Nolan Pokpongkiat, Anna Nguyen.

**Writing – original draft:** Jade Benjamin-Chung.

**Writing – review & editing:** Jade Benjamin-Chung, Benjamin F. Arnold, Chris J. Kennedy, Kunal Mishra, Nolan Pokpongkiat, Anna Nguyen, Wendy Jilek, Kate Holbrook, Erica Pan, Pam D. Kirley, Tanya Libby, Alan E. Hubbard, Arthur Reingold, John M. Colford, Jr.

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
