## [Decision Letter · Decision Letter 0]

28 Jan 2020

Dear Dr. Benjamin-Chung,

Thank you very much for submitting your manuscript "Impact of a city-wide school-located influenza vaccination program over four years on vaccination coverage, school absences, and laboratory-confirmed influenza: a prospective matched cohort study" (PMEDICINE-D-19-04321) for consideration at PLOS Medicine. 

Your paper was discussed among the editorial team, evaluated by an academic editor with relevant expertise, and sent to independent reviewers, including a statistical reviewer. The reviews are appended at the bottom of this email and any accompanying reviewer attachments can be seen via the link below:

[LINK]

In light of these reviews, we will not be able to accept the manuscript for publication in the journal in its current form, but we would like to invite you to submit a revised version that fully addresses the reviewers' and editors' comments. You will appreciate that we cannot make a decision about publication until we have seen the revised manuscript and your response, and we expect to seek re-review by one or more of the reviewers. 

We hope to receive your revised manuscript by Feb 18 2020 11:59PM. Please email us (plosmedicine@plos.org) if you have any questions or concerns.

Please let me know if you have any questions. Otherwise, we look forward to receiving your revised manuscript in due course. 

Sincerely,

Richard Turner PhD, for Louise Gaynor-Brook, MBBS PhD

Associate Editor, PLOS Medicine

rturner@plos.org

To your data statement, please add contact information for the California Emerging Infections Program. 

We ask you to adapt the title so that it is not declarative in style, and suggest: "Evaluation of a city-wide school-located influenza vaccination program in Oakland, California with respect to vaccination coverage, school absences, and laboratory-confirmed influenza: a matched cohort study".

We suspect that your study may best be described as a retrospective analysis of prospectively gathered data, and therefore suggest avoiding claims that the research design is "prospective". 

To your abstract, please add some summary details from table 1 to characterize the intervention and comparison communities (e.g., median household income and ethnicity). 

Where you quote specific findings in your abstract, such as "... the reduction in influenza hospitalizations ... was 76 ... and 165 ...", to assist readers we suggest that you adapt the text to the form "... the reduction ... was 76 individuals ...", for example, and add the size of the relevant group where appropriate. 

Where available, please add p values alongside 95% CI. 

Please revisit your abstract to ensure that the language used (e.g., "intervention ... reduced the incidence" at line 52) is softened to reflect the observational research design (e.g., to "intervention ... was associated with a reduced incidence"), and extend this to the paper's Discussion section, for example. 

After the abstract, we will need to ask you to add a new and accessible "author summary" section in non-identical prose. You may find it helpful to consult one or two recent research papers published in PLOS Medicine to get a sense of the preferred style. 

Please trim the introduction section of your paper. 

We note that you refer to a prespecified analysis plan in the methods section of your main text, and ask that you highlight any non-prespecified analyses. 

Please adapt the early part of your discussion section so that the first paragraph consists mainly of a summary of the study's findings. 

Please adapt journal names where needed (e.g. "PLoS ONE" for reference 69; "U S A" needs to be added to reference 42). Several references are quoted as "in press", which can probably be removed if these are "early online" versions. 

We ask you to add a completed checklist for the most appropriate reporting guideline as a supplementary document (referred to in your methods section), which may be STROBE or RECORD. In the checklist, please refer to individual items by section (e.g., "Methods") and paragraph number rather than by line or page numbers, as the latter generally change in the event of publication. 

Comments from the reviewers:

*** Reviewer #1: 

The manuscript describes the results of a multi-year study to assess the community-wide impacts of a school-located influenza vaccination (SLIV) program that was introduced in the Oakland Unified School District (OUSD) starting in 2014. The study uses many sources of data to estimate various effects of the SLIV program over multiple years with multiple vaccines; this approach would likely appeal to the general readership of this journal. However, I have several major concerns with the manuscript. 

1. It is difficult to follow and keep track of the various endpoints, subgroups, stratified analyses, and sensitivity analyses, especially since neither the schematic of SLIV effects estimated in the study (Figure 1), nor the supplemental material (Appendix S4) include many of the analyses included in the abstract, results, and primary figures. For example, Figure 3 includes estimates among the elderly (not included in Figure 1) but does not include estimates among elementary-aged children (included in Figure 1). The Methods section and accompanying figures should be revised for completeness and clarity. 

2. There is a lack of consistency when referring the various definitions of the influenza season considered in this manuscript that should be resolved, including:

- On page 11, the primary and pre-specified analysis defined the flu season starting when there were two consecutive weeks with %ILI reported to CDPH exceeding a given cutoff. In the primary manuscript, the cutoff of 2.5% was selected after comparing cutoffs of 2%, 2.5%, and 3%, as shown in Appendix S5. However, I do not see mention of alternative cutoffs in the analysis plan. Additionally, in Appendix S10 Fig 13, the caption suggests that the CDC flu season definition is the primary analysis. 

- Is California Emerging Infection Program's influenza season from Oct 1 through April 30 (Page 11, lines 231-232) the same as the period between week 40 and week 20 of each year (Page 13, line 269; Page 14, line 290), and CDC's standard definition (as referred to in Appendix S10 Fig 13)? 

- Finally, the sensitivity analysis for hospitalizations that restricted to the peak week of each influenza season is first introduced in the primary manuscript on Page 18, in the discussion of the results. Since this was prespecified, I would have expected it to be included in the discussion of the methods (it was for absenteeism). 

3. The analyses were designed to estimate incidence and absentee rates, but the authors sometimes report intervention effects in absolute numbers (including in the Abstract). The total numbers aren't interpretable until they are put into a broader context and the total numbers are a simple scaling of the estimates of the rates (Page 14, line 281). I don't mind when the estimated total numbers are presented along with the estimated rates (as in Figure 3), but I am not sure how to interpret results when only the total numbers are presented. Why report only the results in total numbers when the rates are directly comparable?

4. There are several comparisons made, and while p-values are not presented the authors do describe results as statistically significant, or significantly different. How is statistical significance assessed throughout the manuscript? And was there any adjustment for multiple testing? 

5. The claim that the school district catchment areas summarized in Table 1 are "similar" (Page 16, line 323) seems overly broad when there are noticeable differences between the two areas. The differences in median income, percentage of families below the poverty level, and a proportion of the population with college degrees could be associated with the outcomes of interest. If these differences are indicative of systemic difference between the two areas, could these differences confound the estimates of the SLIV program effects? 

6. It looks like the Intervention/Comparison labels at the top of Appendix S3 Table 1 are flipped compared to other summaries of the two school districts, including those in Appendix S6 and Appendix 1 in the pre-specified analysis plan. Please check the column labels. Additionally, based on the description of the matching algorithm on starting on Page 8, line 159, I was expecting Appendix S3 Table 1 to also include summaries of school-level percentages of English language learners and students receiving free lunches.

7. Why are 95% CIs not reported along with vaccination coverage estimates (pages 16-17, Appendix S6)? Since they are reported in the Abstract and described in the Statistical Analysis section (Page 12, line 235), they should be reported throughout the manuscript.

8. The weekly time series in Fig 2 lets us assess whether the trends in the incidence of flu-hospitalization are parallel within a given flu season, but the analyses are done comparing annual flu seasons (so cumulative incidence for the 2015-16 season). How do the weekly (or daily in the case of absenteeism) time series allow us to assess how reasonable the parallel trend assumption is for the DID analyses that are done? And are there similar assessments for the stratified analyses? Lastly, why were school absence trends used to assess whether the equal trend assumption was met for the analyses of influenza hospitalizations (Page 13, line 257)? 

9. Based on the negative control analysis for absenteeism, the authors conclude that differential measurement error may impact the primary analysis (Page 22, line 454). The misclassification analysis gets a lot of attention, but the results of negative control analysis could be explained by many other possible sources of unmeasured confounding. Why was misclassification the only source considered? Could the differences seen in Table 1, along with the results of the negative control analysis be indicative of a poor match for the comparison district? Or could the demographics have shifted differentially in the two districts? 

Minor comments:

- In Table 1, how are the 95% CI's computed for the Median household income? 

- One school dropped out of the vaccine coverage survey, how is that school and its matched pair accounted for in the participant flow in Appendix S9? Where there originally 45 or 46 schools? 

- The categories for the analysis of vaccine coverage by caregiver education level (Appendix S10, Figure 4) and Page 17, line 352 should match than those reported earlier in Appendix S6 and be discussed consistently in the manuscript. 

- Do the estimates in Figure 2 include ICU admissions? Are these total numbers, standardized for potential differences in age groups across the two areas? I would also expect to see these trends for the age subgroups used to obtain the estimates in Figure 3. 

- How was illness-specific absence defined? 

- Does the school-absenteeism analysis use only the matched schools in each district, or all schools in both districts? 

- The caption for Figure 4 should be edited to reflect the definition of flu season that is used. 

- What are the adjusted and unadjusted analyses presented in Appendix S10 Figure 14? And the accompanying caption should clearly define the flu season definitions. 

- The pre-specified plan described the study as a retrospective matched cohort study, but the manuscript refers to the study as prospective. Can you please clarify? 

- How were the estimates of vaccine effectiveness (page 7 and Appendix S2) obtained? 

- The manuscript described power calculations for the vaccine coverage survey but similar computations for the laboratory-confirmed influenza hospitalizations and school absenteeism endpoints (those can be found in the pre-specified analysis plan). These should be organized better.

- I could not find a pre-specified plan for the matching of cohorts or for of the vaccine coverage survey.

- I could not find the code to replicate the analyses online.

*** Reviewer #2: 

In this interesting study, the authors evaluated the impact of school-located influenza vaccination (SLIV) programs implemented in >95 preschools and elementary schools in Oakland, CA during the 2014-15 to 2017-18 influenza seasons. The authors used multiple data sources to examine vaccine coverage, school absences, and laboratory-confirmed influenza hospitalizations. They used a matched cohort design with a comparison community, and used a difference-in-differences (DID) approach to the analyses for hospitalization incidence and absenteeism.

I was impressed by: 1) the inclusion of multiple outcomes determined from independent data sources; 2) the rigor and comprehensiveness of the analyses conducted (with numerous sensitivity and subgroup analyses), and how the authors addressed a number of challenges along the way; 3) the interesting findings of the effects of the program against relatively specific outcomes when differences in vaccine coverage were observed and the vaccine was effective; 4) the appropriate interpretation of the results and the fulsome discussion; and 5) the quality of the writing.

I believe this manuscript would be suitable for publication with just a few minor changes:

1. Abstract Line 30: I believe the surveys should be labelled as 2017 and 2018 rather than 2016 and 2017.

2. Abstract, Lines 45-49: I am wondering if it would be better to report the reductions in outcomes as rates rather than absolute counts. I can appreciate the desire to report the reductions as absolute counts, but would like to hear the authors' rationale for choosing counts over rates.

3. S10 Appendix Table 1: DID in total hospitalizations, Elderly, 2016-17: should be -5 rather than 5. In later rows, the authors should use 2014-15, 2015-16, etc. for the row headings, instead of 1415, 1516, etc. for consistency.

4. Line 406: should be S10 Supplement, not S11 Supplement

Jeff Kwong

*** Reviewer #3: 

GENERAL COMMENTS

This is a very nicely designed study, using aa matched prospective cohort design (matching school districts) and a DID analysis to evaluate the impact on school-located influenza vaccination (SLIV) on: (a) influenza vaccination rates of schoolchildren, (b) school absenteeism, (c) indirect effects (child and adult lab-confirmed hospitalization rates). The study found that during the first two years of SLIV, self-reported vaccination coverage was not higher in intervention schools (when 22-28% of children were vaccinated in school) and flu VE was suboptimal, but in the latter two years (2016/17 and 2017/18) it was higher and there was also a DID difference in the expected direction for school absenteeism and pediatric and adult hospitalizations (same % of children receiving SLIV, but these 2 years the vaccine was more effective). 

The study is novel in that it is methodologically rigorous yet also measures school absenteeism and hospitalizations as indirect outcomes (most prior RCTs of SLIV have not).The study is limited by inherent weaknesses of the design (namely lack of an RCT, self-reported influenza vaccinations); however given the fact that in today's world it is quite unusual to be able to randomize schools and to conduct such large studies within an RCT, the study and findings are important. The authors do a nice job describing the limitations. 

The findings are complicated by multiple co-occurring events or findings—across the 4 years, when similar numbers of students received SLIV vaccinations:

* Overall influenza vaccination rates were lower in intervention schools in Year 1, no different in Year 2, but then much higher in years 3-4 but almost entirely due to vaccination rates in control school districts dropping precipitously rather than vaccination rates rising in intervention schools presumably due to SLIV. This suggests the possibility that it was not SLIV that caused these findings, but rather some unknown drop in vaccination rates in the control school district.

* LAIV (which was given in >85% of SLIV cases) was minimally effective in year 1, not effective in year 2, but IIV which was given exclusively in years 3-4 was effective those years based on national data. So the study findings would predict no impact in years 1-2, but an impact in years 3-4 based merely on national VE data—but only if vaccination rates are indeed higher in intervention vs control school districts.

All of this makes it challenging to follow and understand the findings; but having said that the paper is extremely well written given these constraints.

One key problem I see is self-reported vaccination rates, and particularly a single survey to assess flu vaccinations received over 3 years—the accuracy of this is unclear. Why was CA immunization registry data not assessed to at least verify accuracy? It is possible that recall is better for SLIV districts because parents needed to give consent for vaccination—causing a potential positive (not conservative) bias in ascertainment of vaccination rates. Also the low response rates for the parent surveys, and much lower response rates for the intervention schools (28% vs 38% in control schools) makes these findings a bit more suspect, though I don't have apriori thoughts about the possible direction of bias. 

This is an enormous manuscript; difficult to wade through. I would suggest eliminating race/ethnicity comparisons throughout to make it a bit less cumbersome. I don't see apriori hypotheses for race/ethnicity, and it is confusing since vaccine uptake is lower among black patients and samples get small for the surveys.

SPECIFIC COMMENTS

INTRODUCTION

This section is well written and includes many SLIV-related publications. It also lays the rationale well for this current study- namely that the prior RCTs tended to not study school absences or flu-related illnesses, and the prior studies that did evaluate flu-related hospitalizations or illnesses had somewhat conflicting findings. 

METHODS

SLIV intervention: 

This section is well written. 

Influenza VE:

Although space is of concern, since so much of the findings depend on the national vaccine effectiveness, I suggest either adding the table to the main article (rather than appendix) or stating numbers in the text—eg VE for SLIV in Years 1-2, and for IIV in years 3-4. 

Study Design:

The matching of intervention vs control school districts (cities) was done carefully.

The vaccine coverage survey is problematic since there was one survey that asked parents to recall influenza vaccinations for the prior 3 years (2014-15, 2015-16, 2016-17 flu vaccination seasons), and a second survey that asked about 2017-2018. While prior studies have found good recollection of parental reports of flu vaccinations received in the past vaccination season, I am unsure how accurate self-report would be for 3 seasons. Having said that, there may not be bias in accuracy between intervention and control schools, but there may be bias if the SLIV process (and consent for SLIV) spurred recall of vaccinations received. This is particularly important because the entire study seems to rest upon a DROP in vaccination rates in the control city (school district)- for unclear reasons. 

The analytic approach is otherwise fine. Of note, I did not understand in the method section what differential ascertainment of school absences (measurement error) meant.

RESULTS

 Influenza vaccination coverage:

This section is mostly fine, except I would point out that the difference in coverage between SLIV and control districts was almost entirely due to a DROP in coverage in the control district rather than a rise in coverage in SLIV schools. This lack of a rise in SLIV schools suggests that most SLIV vaccinations were simply substituting for vaccinations otherwise given in doctor's offices. It also brings up the possibility that something unusual happened in the control city to account for the drop in vaccination rates and that the ultimate findings might not be due to SLIV.

Influenza Hospitalization

Hospitalization rates actually ROSE in both cities during the time of a moderately effective vaccine, and the major reason for the DID differences is that they rose markedly in the comparison city in 2016-2017 and 2017-2018 (this is also perhaps related to the marked drop in influenza vaccination rates in that city). 

Absenteeism:

I would add the pre-intervention data to Figure 4, rather than keeping it only in the appendix.

One suggestion is to delete race/ethnicity subgroup analyses in this very extensive manuscript. There is some benefit to keeping race/ethnicity but it could be in a separate manuscript. 

I do not follow the outcome misclassification section when only reading the text. I understand there was a peculiar decrease in illness-specific school absences outside of the flu season, and this is problematic, but the explanation was not sufficient for me to understand what was actually done "under assumed distributions of sensitivity and specificity" or how that reassured the authors that this did not influence the findings sufficiently. 

DISCUSSION

In paragraph 2 the authors state that the vaccination rates for the intervention city was higher than rates for the control city because the latter "did not recover from the decline associated with the switch from LAIV to IIV vaccines." This might be true, but nationally vaccination rates did not drop more than 1-2 percentage points during this time, and the authors do not show data to substantiate this claim. For example, how much LAIV was used in the control city prior to its termination? Is this all speculation or are there data? Does CAIR provide any answers—ie what was the city-wide vaccination rates in CAIR before and after this time period? I am worried that something else, unmeasured, may be going on.

The paragraph on potential substitution is fine, but perhaps over-calling the impact of SLIV, since it implies that without SLIV the coverage rate in the intervention city of Oakland would have dropped markedly. This did not occur nationally (flu vaccination rates for 5-12yr olds dropped by 2 percentage points between 2015/16 and 2016/17) or I believe statewide across California—so why would one expect it to drop so much?

Overall the rest of the discussion section was well done.

***

[LINK]

---

## [Decision Letter · Decision Letter 1]

12 Jun 2020

Dear Dr. Benjamin-Chung,

Thank you very much for re-submitting your manuscript "Evaluation of a city-wide school-located influenza vaccination program in Oakland, California with respect to vaccination coverage, school absences, and laboratory-confirmed influenza: a matched cohort study " (PMEDICINE-D-19-04321R1) for review by PLOS Medicine.

I have discussed the paper with my colleagues and the academic editor and it was also seen again by xxx reviewers. I am pleased to say that provided the remaining editorial and production issues are dealt with we are planning to accept the paper for publication in the journal.

[LINK]

We look forward to receiving the revised manuscript by Jun 19 2020 11:59PM. 

Sincerely,

Thomas McBride, PhD

Senior Editor 

PLOS Medicine

plosmedicine.org

Requests from Editors:

1- Thank you for editing the Discussion for causal language. Please also edit the first sentence of the Abstract Conclusions: “A city-wide SLIV intervention in a large, diverse urban population *was associated with a decrease in* the incidence of laboratory-confirmed influenza hospitalization in all age groups and *a decrease in* illness-specific school absence rates among students during seasons... ”

2- Similarly, in the Discussion Conclusions: “Offering school-located influenza vaccination to all elementary schools in a large, urban district led to 7-11% *was associated with* increases in vaccination, which *was followed by* meaningful reductions in community-wide reductions in influenza hospitalization and illness-specific absences among school children among those not targeted by the program, including the elderly.”

3- Introduction, beginning of the 3rd paragraph: “...as *a* strategy…”

4- Introduction, end of the 3rd paragraph, please edit to read: “To our knowledge, no prior studies….” or similar.

5- Thank you for noting that documented informed consent was waived, please specify that it was the Committee for the Protection of Human Subjects that provided this waiver (I’m assuming that’s who provided the waiver).

6- Please also move the Ethical Statement towards the beginning of the Methods (so it appears as the first or second section).

7- Thank you for providing the completed STROBE checklist. Please add the following statement, or similar, to the Methods: "This study is reported as per the Strengthening the Reporting of Observational Studies in Epidemiology (STROBE) guideline (S_ Checklist)."

8- When presenting the demographic information in the Results section, please include the 95%CIs.

9- Please adjust the wording at lines 57, 522 and 623 to more clearly reflect that the reduction in admission was not significant until years 3 and 4. 

10- Please add the STROBE checklist to the list of Supporting Information items at the end of the main text.

11- Please use the "Vancouver" style for reference formatting, and see our website for other reference guidelines https://journals.plos.org/plosmedicine/s/submission-guidelines#loc-references

Comments from Reviewers:

Reviewer #1: The authors have done a great job revising the manuscript to address the reviewer comments and I appreciate the more streamlined manuscript. I think this manuscript is suitable for publication, however I do have one minor comment:

- The parallel trend assumption for the DID analyses is untestable and I would suggest the authors soften the language to something like "the data indicates the assumption of equal trends is reasonable", rather than "equal trends assumption was met" (lines 295 and 324 for example). 

Reviewer #2: The authors have addressed my comments. I have no further comments.

Reviewer #3: This is a very nicely designed study, using a matched prospective cohort design (matching school districts) and a DID analysis to evaluate the impact on school-located influenza vaccination (SLIV) on: (a) influenza vaccination rates of schoolchildren, (b) school absenteeism, (c) indirect effects (child and adult lab-confirmed hospitalization rates). The study is novel particularly in assessing "b" and "c" since more rigorous studies without the limitations below have assessed "a". 

The authors dis a very nice job responding to the many critiques. The two major critiques I had included (1) concern that their study rests on a drop in self-reported influenza vaccination coverage among comparison schools rather than an increase in SLIV schools and (2) self-reported influenza vaccination without any validation via CAIR or other methods. 

The authors responded to #1 by highlighting the large % of vaccines given via LAIV in 2014-15 and 2015-16 (prior to it being discontinued) and the drop in vaccination rate in comparison schools being approximately the same as the proportion that had been given via LAIV. This is plausible. Apparently CAIR was not sufficiently complete to be used as a validation for self-reported vaccination rates, and it is difficult to see why there would be such a large difference in the bias in self-reported rates for comparison vs control schools. This is a limitation, but I feel it likely did not affect the results substantially.

Overall this is a very nice paper and revision.

[LINK]

---

## [Editor Report · Decision Letter 2]

14 Jul 2020

Dear Dr. Benjamin-Chung, 

On behalf of my colleagues and the academic editor, Dr. Mirjam Kretzschmar, I am delighted to inform you that your manuscript entitled "Evaluation of a city-wide school-located influenza vaccination program in Oakland, California with respect to vaccination coverage, school absences, and laboratory-confirmed influenza: a matched cohort study " (PMEDICINE-D-19-04321R2) has been accepted for publication in PLOS Medicine. 

PRODUCTION PROCESS

PRESS

PROFILE INFORMATION

Thank you again for submitting the manuscript to PLOS Medicine. We look forward to publishing it. 

Best wishes, 

Thomas McBride, PhD

Senior Editor 

PLOS Medicine

plosmedicine.org